# VEGFA, B, C: Implications of the C-Terminal Sequence Variations for the Interaction with Neuropilins

**DOI:** 10.3390/biom12030372

**Published:** 2022-02-26

**Authors:** Charles Eldrid, Mire Zloh, Constantina Fotinou, Tamas Yelland, Lefan Yu, Filipa Mota, David L. Selwood, Snezana Djordjevic

**Affiliations:** 1Structural and Molecular Biology, ISMB, Division of Biosciences, University College London, Gower Street, London WC1E 6BT, UK; c.eldrid@soton.ac.uk (C.E.); ntina.fotinou@gmail.com (C.F.); tamas.yelland06@evotec.com (T.Y.); lefan.yu.17@alumni.ucl.ac.uk (L.Y.); 2UCL School of Pharmacy, University College London, London WC1N 1AX, UK; zloh@live.co.uk; 3Faculty of Pharmacy, University Business Academy, 2100 Novi Sad, Serbia; 4Wolfson Institute for Biomedical Research, University College London, Gower Street, London WC1E 6BT, UK; filipamotapt@gmail.com (F.M.); d.selwood@ucl.ac.uk (D.L.S.)

**Keywords:** X-ray crystallography, ligand-binding protein, neuropilin, SPR, molecular dynamics

## Abstract

Vascular endothelial growth factors (VEGFs) are the key regulators of blood and lymphatic vessels’ formation and function. Each of the proteins from the homologous family VEGFA, VEGFB, VEGFC and VEGFD employs a core cysteine-knot structural domain for the specific interaction with one or more of the cognate tyrosine kinase receptors. Additional diversity is exhibited by the involvement of neuropilins–transmembrane co-receptors, whose b1 domain contains the binding site for the C-terminal sequence of VEGFs. Although all relevant isoforms of VEGFs that interact with neuropilins contain the required C-terminal Arg residue, there is selectivity of neuropilins and VEGF receptors for the VEGF proteins, which is reflected in the physiological roles that they mediate. To decipher the contribution made by the C-terminal sequences of the individual VEGF proteins to that functional differentiation, we determined structures of molecular complexes of neuropilins and VEGF-derived peptides and examined binding interactions for all neuropilin-VEGF pairs experimentally and computationally. While X-ray crystal structures and ligand-binding experiments highlighted similarities between the ligands, the molecular dynamics simulations uncovered conformational preferences of VEGF-derived peptides beyond the C-terminal arginine that contribute to the ligand selectivity of neuropilins. The implications for the design of the selective antagonists of neuropilins’ functions are discussed.

## 1. Introduction

Neuropilin-1 (NRP1) and neuropilin-2 (NRP2) are single-pass transmembrane proteins with approximately 840 amino acid residues in their extracellular (ectopic) domain and a small intracellular region containing 44 residues [1,2]. While the ectopic region, comprising five structural domains, interacts with a range of physiological ligands, the intracellular region is devoid of any catalytic activity, and although studies show distinct roles for the cytoplasmic peptide [3,4], neuropilins are not viewed as canonical functional receptors. Instead, the role of these proteins is commonly described as that of co-receptors, since many of their physiological functions are tightly linked to supporting the activities of other well-characterised receptors, such as several transmembrane tyrosine kinases and plexins [1,5]. Since their discovery as the receptors for class 3 semaphorins, neuropilins were shown to be involved in binding a range of other ligands including vascular endothelial growth factors, TGF-beta, hepatocyte growth factor, galectins and several viral proteins [5,6,7,8,9].

Studies of neuropilins’ function in mediating responses to vascular endothelial growth factors (VEGFs) dominate the literature. In mammals there are four VEGF genes designated VEGFA, VEGFB, VEGFC and VEGFD that code for structurally homologous proteins [10]. All proteins share a key structural domain, known as a cysteine knot, which is recognised by and interacts with the corresponding VEGF receptors (VEGF-Rs). Apart from the common cysteine knot domain, the proteins display significant diversity either because of alternative splicing or due to posttranslational proteolytic processing. Structural differences are paralleled by the functional specialisation with respect to the receptors with which they engage and the processes that are regulated by these interactions. Furthermore, neuropilins are required to reach the full activity of various VEGF-VEGF-R signaling pathways, adding an additional layer of complexity.

VEGFA is a key regulator of vasculogenesis and angiogenesis, and it signals through interactions with VEGF-R2 and NRP1. VEGFC activity, mediated via NRP2/VEGF-R2 (or VEGF-R3) receptor complexes, plays a critical role in lymphangiogenesis [11]. While VEGFC and VEGFD have partially overlapping functionalities, VEGFC is essential, as its genetic deletion leads to the lethality of knockout mice, while VEGFD is not [12,13]. VEGFB, which is most prominently expressed in cardiac and skeletal muscle [14], selectively binds to VEGF-R1 [15] and NRP1 [16] and has been implicated in diabetes [17,18]. VEGFB was also identified as an antioxidant upregulating a cluster of antioxidant proteins [19].

How is the regulation of these varied VEGF functions achieved? A degree of regulation is likely attained by the levels of expression of the specific genes within different tissues. However, the overlapping presence of the receptors and the ligands raises the possibility for differentiation through the specificity of the molecular interactions. The key aspect of the interaction between neuropilins and VEGFs occurs through binding of the C-terminal arginine of VEGF to the ligand-binding region of neuropilin b1 domain [20,21]. The interactions between neuropilins and VEGFs have been previously studied using X-ray crystallography. Specifically, the interactions of VEGFC with NRP2 were described based on the structure of a heterocomplex between the NRP2-b1 domain and the C-terminal sub-domain of VEGFA [22] and in the context of a crystal structure of fusion proteins when the C-terminal pentapeptide of the mature VEGFC was fused to the C-terminus of the NRP2-b1b2 domain’s protein construct [23]. Similarly, for NRP1, a crystal structure of the fusion protein where the VEGFA heparin-binding domain was fused to the C-terminus of the NRP1-b1 domain has been reported [24]. Importantly, the b1 domain of neuropilins was identified as the binding site not only for VEGFs, but also for other natural and synthetic ligands that contain the R/K*XX*R sequence motif that follows the so-called “C-end rule” and that is commonly generated by the action of furin protease [25]. It was recently recognised that furin protease activity is uniquely required for the activation and attachment of the SARS-CoV-2 spike protein, contributing to the effectiveness of SARS-CoV-2 viral infection and further raising interest in neuropilins expression, function and the molecular basis of ligand binding [9].

We have carried out a comprehensive analysis of the interactions between NRP1 and NRP2 with the pentapeptides corresponding to the C-termini of the four physiological ligands from the VEGF family: VEGFA165, VEGFB167, VEGFB186 and VEGFC (identical in this region to VEGFD), as well as the control peptide with an AAAAR sequence. The length of peptide sequences was selected to correspond to the length of the C-terminal sequence of the key physiological molecule VEGFA165. This sequence, encoded by Exon 8, extends from the globular region of the heparin-binding domain. We report three crystal structures with peptide complexes, which were then used as the models to generate all possible molecular complexes and to carry out molecular dynamics simulations. We correlated our findings with surface plasmon resonance (SPR)-determined binding affinities and conclude that physiological activities are not a reflection of significant differences in binding affinities for the ligands from the VEGF family; instead, we conclude, the ligands’ selectivity might be directed by the conformational preferences beyond the C-terminal arginine and transient protein-protein interactions that are formed with neuropilins, as observed in molecular dynamics simulations.

## 2. Materials and Methods

### 2.1. Protein Expression and Purification

Proteins were expressed and purified as previously described [22,26]. NRP1-b1 and NRP2-b1 were expressed in Rosetta-gami2(DE3)pLysS *E.coli* cells that were transformed with a pET-15b vector containing NRP1-b1 and NRP2-b1 sequences with an N-terminal 6-His-tag and a TEV protease cleavage site. After the cells were harvested by centrifugation and lysed by sonication, the proteins were purified from the filtered cell extract. Purification protocol, as described in Tsai et al. (2016) [22], included a combination of nickel ion-affinity chromatography, size exclusion, and a final ion exchange chromatography step.

### 2.2. Peptides

All peptides (Table 1) were produced as the N-acetylated form by PeptideSynthetics, Fareham, UK and were used without further purification. In the manuscript the pentapeptides are referred to by isoform number, i.e. ‘A165′ refers to VEGFA165 pentapeptide, etc. Control peptide is labeled as ‘Con’ and ‘Ac’ is used to denote Acetyl-group.

### 2.3. Surface Plasmon Resonance

SPR was performed using the Biacore T200 Enhanced Sensitivity model (GE, Healthcare, UK). Protein was immobilised on a CM5 chip, following the manufacturer’s instructions until it reached 1000 response units (RU). SPR was run with a range of peptide concentrations from 64 µM to 0.125 µM, in a buffer of 20 mM Tris pH 7.9, 50 mM NaCl. Biacore T200 evaluation software was used for data analysis.

### 2.4. Crystallography

Initial crystallisation screens were performed using the Qiagen PEGs II Suite of 96 conditions via the sitting drop vapour diffusion method and pipetted by a TTP Labtech Mosquito. 100 nL of protein solution was mixed with 100 nL of reservoir solution in each drop. Optimisation screens were performed manually, through hanging drop vapour diffusion, with 1 µL of protein solution mixed with 1 µL of reservoir solutions. Suspension of crystallisation seeds was produced using the Hampton Research Seed Bead kit. In crystallisation experiments using seeds, 0.5 µL of seeds suspension were added to an existing 2 µL protein/reservoir solution drop. Final crystallization conditions were as follows: NRP1/B186–0.2 M (NH_4_)_2_SO_4_, 0.1 M MES pH 6.5, 30% (*w/v*) PEG5000 MME; NRP2-b1/B167-0.1 M HEPES pH 7.5, 10% (*w/v*) PEG4000, 20% (*w/v*) isopropanol; NRP2-b1/C-0.1 M NaOAc, 0.1 M HEPES pH 7.5, 12% (*w/v*) PEG4000. Crystals were transferred into cryoprotectant solution comprising the reservoir solution enriched with 20% Ethylene Glycol and stored in liquid nitrogen prior to data collection. Data collection was performed at the Diamond Light Source synchrotron beamlines I03 and I04. X-ray diffraction data processing was performed using Xia2 3dii and auto-PROC [27,28]. Molecular replacement structure solution was performed using Phaser MR [29] with 4RN5 (PDB ID) as a starting model. Crystal structures were built in Coot [30] and structural refinement was performed using Phenix [31] software. Structural figures were produced in Pymol.

### 2.5. Thermostability Shift Assays

Thermostability assays were performed on the LightCycler 480 II (Roche) with a final concentration of 5 µM protein, 1 mM ligand and 10X of SYPRO Orange, in DMSO (Thermo Fisher). Negative and positive control experiments were performed using the protein with buffer alone or the ligand without protein and the protein with a strongly binding ligand EG01377, respectively. Experiments were carried out in duplicates on two separate days for two different batches of protein preparations and the values in the table represent the averages from the four measurements.

### 2.6. Molecular Dynamics Simulations

Molecular dynamics simulations of NRP1 and NRP2 proteins in complex with peptides were conducted using Desmond molecular dynamics software v3.6.0 [32,33] and the OPLS2005 force field [34]. The three-dimensional structures of protein peptide complexes obtained from X-ray crystallography were imported into Maestro. The structures were prepared for simulations using Protein Preparation Wizard by setting protonation states of ionizable groups to mimic the effects of environment at pH 7 and adding hydrogen atoms. The System Builder module of Desmond software was used to fully solvate all systems with explicit water using a simple point charge (SPC) model and 50 mM NaCl. The size of the periodic boundary conditions box was set to allow a minimum 10 Å distance from the protein-peptide system to the nearest side of the box. Each system was equilibrated using the default simulated annealing protocol by raising temperature with a linear gradient between the following set points: 0 K at start; 10 K at 30 ps; 100 K at 100 ps and finally 1000 K at 1000 ps. The NVT ensemble class (simulation at the constant volume) was used with a Berendsen thermostat (1 ps relaxation time). The RESPA integrator was set to 1 fs and near = 1 and far = 3. The coulombic interaction cut-off was set to 9 Å. The time step was set to 1 fs, the energy recording interval 1 ps and the trajectory interval 5 ps. The final frame of the simulated annealing trajectories for both structures were subjected to the production simulation at 300 K for 10 ns using the same setting as for simulated annealing except for the NPT ensemble class (simulation at the constant pressure). The final frame of trajectories for these two structures were used to generate the protein-peptide complexes for which the X-ray structures were not available. Water molecules and ions were deleted from both systems and the peptide sequences were mutated accordingly. The System Builder was used to solvate complexes and add ions. All complexes were submitted to 10 ns equilibration simulations 300 K and constant pressure, as described above. Final frames extracted from the equilibration trajectories were used as starting points for 30 ns production simulations. All production simulations were run in duplicate. Trajectories were analysed using tools and scripts implemented in Maestro, including the interactions of peptides with proteins by Simulation Interaction Diagram tool.

## 3. Results

### 3.1. Interaction with Peptides Increases Thermostability of b1 Domains

The peptides used in this study were based on the C-terminal five residues of human isoforms of VEGF, specifically VEGFA165, VEGFB167, VEGFB186, VEGFC and a control peptide, as shown in Table 1. The pentapeptides are referred to by isoform number, i.e., ‘A165′ refers to VEGFA165 pentapeptide, etc. The control peptide is labelled as ‘Con’. All peptides were used to carry out initial comparisons of interactions with the b1 domains of NRP1 and NRP2 using thermostability shift assays. These assays are commonly used in screening for small molecules’ compounds, which will bind to the protein of choice, resulting in a ligand-induced stabilisation of the protein and an increase in the corresponding apparent melting temperature (Tm). We have previously demonstrated that NRP1-b1 domain exhibits a one-stage unfolding mechanism with a single Tm, whereas NRP2-b1 protein involves a two-stage unfolding mechanism with two corresponding Tm values [22]. All peptides in this study caused shifts in the melting temperatures of NRP1-b1 and NRP2-b1, thereby confirming their binding to the b1 domain. The stabilising effect of some peptides is stronger than others (Appendix A) and none of the peptides influenced the Tm2 of NRP2-b1. For NRP1-b1, the largest shifts in Tm of 3.5 ± 0.2 °C and 3.6 ± 0.2 °C were detected in the presence of A165 and B186, respectively. The control peptide caused a similar shift in Tm of 3.0 ± 0.2 °C, in agreement with the presence of the C-terminal arginine (Table 1) being sufficient for interaction with NRP1-b1. Consistent with the physiological selectivity of NRP1, the lowest thermal stability effect was observed in the presence of peptide C (ΔTm = 2.0 ± 0.3 °C). In contrast, with respect to the NRP2-b1 domain, peptides A165, B167 and C induce an increase in Tm1 by approximately 2.4 °C while B186 has the most stabilising effect, raising Tm1 by 3.5 ± 0.2 °C.

### 3.2. All Peptides Exhibit Binding Affinities in the Micromolar Range by SPR

The SPR data exhibited fast on and off binding kinetics with the Kd values in the low micromolar range (Table 2 and Appendix A). Determined affinities range from 4 μM for B167 binding to NRP1 to >90 μM dissociation constant for the interaction of the control peptide with NRP2. Interestingly, NRP1-b1 showed both higher selectivity between the peptides and higher affinity for the peptides compared to NRP2-b1. The poorer selectivity of NRP2-b1 for the peptides, observed in the SPR analysis, mirrored the effect that these peptides, except B186, had on Tm1 values for NRP2. NRP1, which is commonly considered to be the primary receptor for VEGFA, exhibited the weakest affinity for VEGFC-derived peptide and the control peptide. In contrast, NRP2, the main physiological receptor for VEGFC, did not differentiate significantly between VEGFA and VEGFC-derived peptides and interacted even more poorly with the control peptide. For both NRP1 and NRP2, B167, the most positively charged peptide of those examined, exhibited the highest affinity in SPR essays.

### 3.3. X-ray Crystal Structures of the Complexes Exhibit Distinct Features

While preliminary crystals were generated for several neuropilin/peptide combinations, good quality data were collected and the structures were solved and refined for the following three new complexes: NRP1-b1/B186, NRP2-b1/B167, and NRP2-b1/C. These crystals were obtained by a microseeding method with seeds generated from native neuropilin crystals and reservoir solutions containing the specific buffer combinations indicated in the Materials and Methods section. The three complexes formed crystals that exhibited P2_1_2_1_2_1_, C2 and P2_1_ symmetry, respectively. X-ray data and structure refinement statistics are presented in Table 3. The structures were deposited to the PDB with accession numbers PDB ID: 6TKK, 6TDB, 6TJT.

In all three structures, the peptides are bound within the canonical ligand-binding site in the b1 domain. Neuropilin b1 domain belongs to the structural family of coagulation factor V/VIII homology domains, also known as discoidin domains, and the ligand-binding site is formed by three loops labelled L1–L3, which are projecting from the two central β-sheets. The C-terminal arginine residue from the ligand peptides is clearly observed in the electron density maps forming interactions with the residues Y297/Y299, S346/S349, D320/D323, T349/T352 and Y353/Y356 in NRP1/NRP2 (Figure 1). Equivalent interactions have been described in the previously reported ligand-bound structures and they are primarily associated with the binding of the carboxylate group and the guanidinium group of the C-terminal arginine within the specific binding site [22,23,24].

Apart from the commonalities of interactions between the C-terminal arginine of the peptidic ligands and the b1 domain ligand binding site [22,23,24,35], the structures showed differences in the ways the peptides interacted with the rest of the structure. The structure of the NRP2/C complex contained two molecular complexes in the asymmetric unit. Each of the NRP2 molecules had a VEGFC-derived peptide bound, but we were able to confidently place only the three C-terminal residues, while the two N-terminal residues of the pentapeptide were disordered (Figure 1, middle). The structure of the NRP1/B186 complex, which contained a single NRP1-b1 domain in the asymmetric unit, had a clear and unambiguous high-resolution electron density for all five residues of the peptide (Figure 1, left). Inspection of the structure revealed that the presence of the well-ordered peptide is most likely a consequence of the peptide’s interaction with the symmetry-related neuropilin molecule. Namely, backbone atoms of the B186 peptide formed hydrogen bonds with the backbone atoms of the symmetry-related NRP1 residues within the loop L2 (residues 320–322).

The asymmetric unit of the crystal structure of the NRP2/B167 complex contains four NRP2-b1 molecules, but only three B167 peptides were identified (Figure 2A). Two of the NRP2 ligand binding sites contained the peptides in a canonical-binding mode, again with only 2 to 3 C-terminal residues identifiable within the electron density map while the remainder of the peptide displayed disorder and was not modeled. The third peptide molecule was fully modelled in a well-defined electron density that covered 4 of the residues while the electron density for the 5th, N-terminally acetylated residue, displayed some disorder. Despite this disorder, it was clear that the peptide, represented in Figure 2B, reaches out to the b1 domain ligand-binding area on a neighbouring NRP2-b1 chain, thus creating a bridge between two of the four NRP2-b1 molecules in the asymmetric unit. This was an unexpected observation; however, the peptide B167 contains an almost palindromic sequence, RKLRR, and the presence of the carbonyl oxygen atom from the acetyl group at the N-terminal arginine introduced a hydrogen bond acceptor atom that could engage in an interaction with the ligand-binding site on the NRP2-b1 domain (Figure 2B). Positioning of this N-Acetylated, N-terminal Arg residue departs from the canonical C-terminal Arg-binding mode but bears similarity to what we have previously observed in the structure of NRP1-b1 domain complexed with L-homoarginine (PDBID: 5IJR). The N-acetyl group is oriented towards Trp304, with the carbonyl oxygen thereby forming a hydrogen bond only with Ser349 and losing the interaction with Thr352 (Thr349 in NRP1), which is otherwise one of the hallmarks of the canonical binding mode for the C-terminal arginine, as described above [22,23,24,35]. We also note that in the structure of the NRP2-b1/B167 complex, the C-terminal arginine of the peptide formed an ionic interaction with only one of the D323 carboxylate atoms. Moreover, in the molecule of the NRP2-b1/B167 complex, in which the peptide was fully modelled (Figure 1, right), Arg4 interacts with D301, but this was not seen for the other two peptides in the asymmetric unit.

### 3.4. MD Simulations of Molecular Complexes Reveal Distinct Patterns of Sampling of Conformational Space by the Peptidic Ligands

To further investigate the specific aspects of neuropilins’ interactions with their VEGF ligands, we carried out MD simulations on all NRP/VEGF peptides combinations, including a putative complex with the control peptide. Figure 3 illustrates the results of simulations of complexes involving the NRP1-b1 domain. Centroids of the most populated clusters were extracted from trajectories of 30 ns molecular dynamics simulations and the corresponding peptides were shown as solvent-accessible surfaces laid over the NRP1-b1 domain, displayed as a grey ribbon. In all images, Glu348 (shown in CPK rendition), the residue positioned on the top of the ligand-binding site, was used as a reference point. This representation of MD simulation results illustrates the space occupied by various peptides, indicating that although they all share the interactions related to the tethering of the C-terminal arginine to the NRP1-b1 domain, the flexibility of the peptides and the space which they occupy beyond the binding pocket vary significantly. This is further illustrated in Figure 4, which depicts the extent of the conformational space sampled by VEGF peptides. The two figures show that compared to the other peptides, the representative conformations from the most occupied clusters for the VEGFA165-derived peptide occupy a narrower conformational space (Figure 4), which is reflected by a lower dispersion of the backbones for cluster representatives. In addition, the structures are positioned closer to the surface of the NRP1-b1 molecule, as shown in Figure 3 by the orientation of the centroid of the most populated cluster. Data were further analysed in terms of specific contacts that the peptides were making with the NRP1-b1 residues and expressed as a fraction time of interaction occurrence (Table 4). This analysis effectively indicates how often the specific neuropilin residue came into contact with the peptide during the 30 ns of simulations. Table 4 and Appendix A demonstrate differences in patterns of interactions between peptides and NRP1-b1. For example, the control peptide interacts with fewer NRP1 residues than any of the VEGF-derived peptides; the B167 peptide also forms fewer interactions compared to A165, B186 and C despite its strong preference for Glu348, as shown by the 93% time-of-interaction occurrence with this residue. Neither of the two VEGFB-derived peptides exhibits significant interaction with Thr349. In these simulations, interestingly, neither peptide B167 nor peptide C formed significant interactions with Tyr353. In all crystal structures of molecular complexes of NRP1 with its natural ligands, as exemplified in Figure 1A, Tyr353 was identified as one of the key residues forming a hydrogen bond with the carboxylate oxygen of the ligand’s terminal arginine while simultaneously engaged in stacking interactions with the aliphatic portion of the arginine side chain. Loss of these interactions in the MD simulations would be consistent with the lack of preference of NRP1 for VEGFC.

The same type of analysis was carried out for simulations involving the NRP2-b1 domain and is displayed in Figure 5 and Figure 6 and Table 5 (Appendix A). Again, a pattern emerges where the peptides could be differentiated based on their preferred orientation. Table 5 clearly shows that in the case of NRP2-b1 domain, the control peptide also exhibits fewer interactions overall compared to the other peptides. However, in contrast to what was observed for the NRP1-b1 domain, it does engage in interactions with T352 (T349 in NRP1), but only 60% of the time, similarly to what is seen for A165. Another interesting feature that the simulations have uncovered is that for the 30 ns simulation, peptide C corresponding to VEGFC, the main physiological substrate of NRP2, interacts with Arg303, while this interaction is absent for A165 and B167, which, in contrast, interact with Asp301 through a water-mediated hydrogen bond. Arg303 and Asp301 are positioned within the L1 loop of the ligand-binding domain of neuropilins. This loop contains the most structurally divergent sequence region of b1 domain when comparing the amino acid sequences of NRP1 and NRP2. In NRP2, the L1 loop (YSDGRW) is one residue longer than that of NRP1 (YSTNW). While the MD simulations uncovered the potential of Arg303 to interact with the carboxyl group from the VEGF-C-derived peptide (Appendix A), in the crystal structures of NRP2, Arg303 is engaged in an internal ionic interaction with Asp315, which is maintained even when a peptidic ligand is present. However, our findings from MD simulations are consistent with the previously reported mutagenesis study, where the L1 loop was identified as a major contributor to NRP1/NRP2 selectivity for VEGFA [36].

## 4. Discussion

We examined a series of peptides corresponding to the C-terminal ends of the subset of physiological ligands for neuropilins. Specifically, we looked at the family of vascular endothelial growth factors which are involved in the formation, remodelling and function of the vasculature. While these ligands mediate their effects through interaction with VEGF receptors, the full activity and regulation of downstream processes requires the involvement of neuropilins. How the ternary complexes are assembled and how the physiological specificity is achieved is still not fully described. Here we examined the contribution to these processes made by the C-terminal binding sequences of the VEGF isoforms.

Several molecular mechanisms are usually involved in the control of the receptor tyrosine kinase activity such that interaction with the related ligands could be reflected in the activation of the distinct downstream signalling pathways and different cellular outcomes [37]. In the case of VEGFs/VEGF-Rs, we propose that the biased agonism of various VEGFs is enabled by neuropilins, acting as the non-catalytic coreceptors that stabilise the specific ligand/receptor interactions and the associated conformational state.

Crystals structures of the complexes between the b1 domain and the VEGF-derived peptides shown here exhibit only minor differences in their binding modes with only the most terminal residues of the peptides providing a significant contribution to the interactions with the neuropilin b1 domain irrespective of the peptide or the neuropilin. The structures further indicate that the remaining residues within the C-terminal regions are flexible and disordered, at least in the context of examined peptides. The values of the SPR-determined binding constants support these observations, revealing only minor differences in the strength of the interactions and suggesting that the physiological selectivity and ligand preferences of NRP1, compared to NRP2, is achieved through more complex mechanisms.

We employed MD simulations to complement our experimental approaches and uncovered a distinct pattern of conformational clusters that the VEGF-derived peptides populate upon the C-terminal arginine-mediated interaction with the b1 domain binding site. The data are consistent with the mechanism where the b1 domain provides a tethering point for the multiple, partially overlapping binding interfaces for these ligands. Within the full-length VEGFA165 and VEGFB167 proteins, the five C-terminal residues, expressed by Exon 8, are linked to the core cysteine knot domain that interacts with the VEGF-Rs via the so-called heparin-binding domains, expressed by Exon 7, and comprising about 50 residues (Figure 7A). In contrast, VEGFB186 and VEGFC are products of proteolytic processing by furin and in those proteins, the corresponding five C-terminal residues are linked to the core cysteine knot domains only via several-residues long linker, without the intervening heparin-binding domains (Figure 7A). Assembly of the specific ternary complex between one of the VEGF receptors (R1, R2 or R3) and one of the neuropilins (NRP1 or NRP2) together with the specific VEGF family member (A, B, C) would spatially restrain the orientation of the bridging peptide. The preferential interaction of different peptides with the surface of the neuropilins (Figure 7B) could be a contributor to selectivity through stabilisation of the conformational assembly competent in triggering the corresponding signalling response. In addition, the presence of a heparin-binding domain in VEGFA165 and VEGFB167 indicates that these proteins also interact with the heparan sulfate component of the extracellular matrix [38], further restricting the organization of the receptors/VEGF complexes on a cell membrane. This molecular mechanism not only explains the physiological selectivity between the members of the VEGF family but also supports the notion that selective inhibition of neuropilins and VEGF activity would require an understanding of the protein/protein interactions in the context of ternary signalling complexes. The work described here also illustrates the power of combining MD simulations with experimental binding data and structural analysis.

## Figures and Tables

**Figure 1 biomolecules-12-00372-f001:**
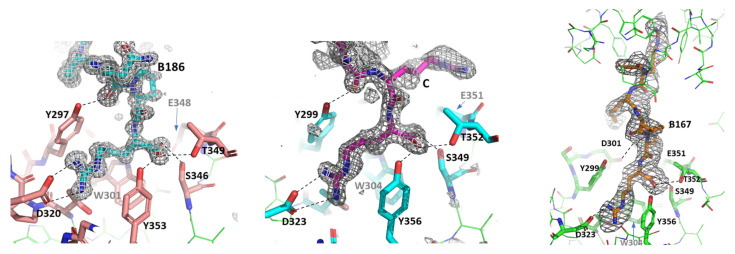
Illustration of the quality of the electron density maps within the ligand-binding sites of NRP1-b1/B186 (**left**), NRP2-b1/C (**centre panel**) and NRP2-b1/B167 complexes (**right panel**). Fo-Fc omit maps were calculated for the peptide ligands and displayed at 3.0, 1.5 and 1.5 σ levels for B186, C and B167 peptides, respectively. In each of the panels, the main residues forming the ligand-binding site are labelled and the hydrogen bonds between the ligand and the protein side chains are indicated by dashed lines. For clarity, water molecules are excluded from these images. Note that in the crystal structure of the NRP1-b1/B167 complex, the C-terminal arginine of the peptide forms hydrogen bonds only with one of the D323 atoms. In the structure of the NRP2-b1/B167 complex, one of the peptides bridges two molecules of NRP2-b1 protein and the electron density for the entire peptide is shown.

**Figure 2 biomolecules-12-00372-f002:**
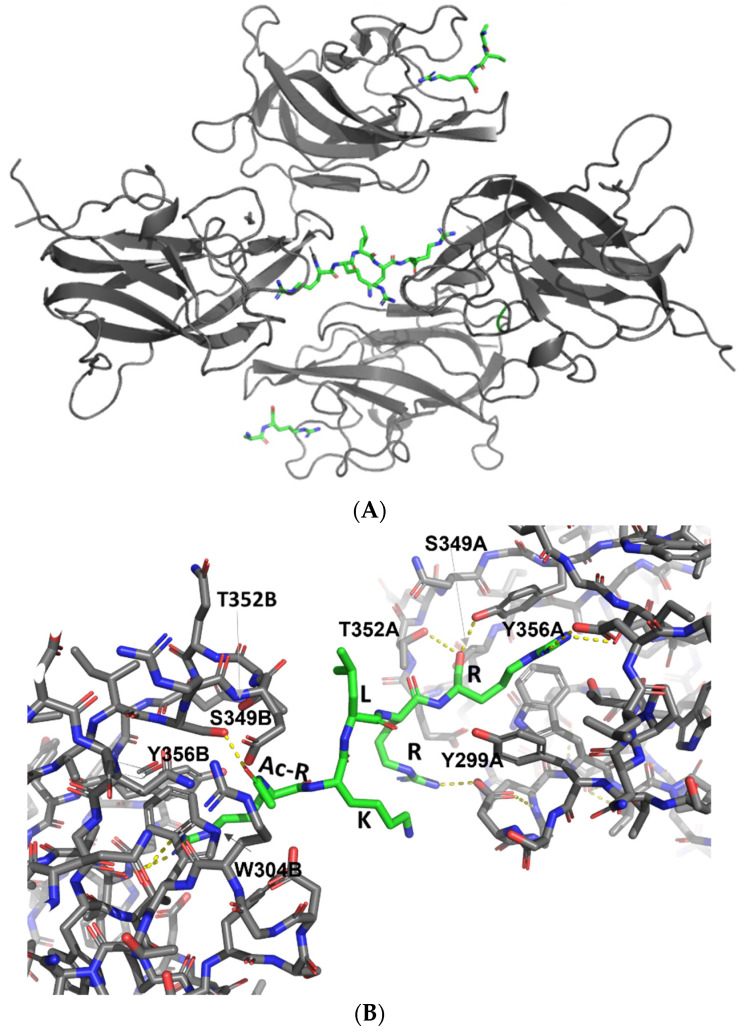
(**A**). The content of the asymmetric unit within the structure of NRP2-b1 domain in a complex with B167 peptide. Four molecules are shown as grey ribbons while the peptidic ligands are represented as sticks coloured by atoms. The two peptides with only two/three modeled amino acids are at the top and the bottom, whereas the peptide with five modeled amino acids (represented in Figure 1, right) is in the center. (**B**). Close-up view of interactions between B167 peptide and the two neuropilin chains within the crystal structure. The residues within the peptide sequence RKLRR are labeled and the N-terminal acetylated arginine of the peptide is indicated by Ac-R. The structures are shown as sticks with the neuropilin carbon atoms within chains A and B coloured grey while the carbon atoms of B167 peptide are coloured green. All nitrogen and oxygen atoms in the figure are coloured blue and red, respectively. Hydrogen bonds formed by the peptide are denoted by dashed yellow lines.

**Figure 3 biomolecules-12-00372-f003:**
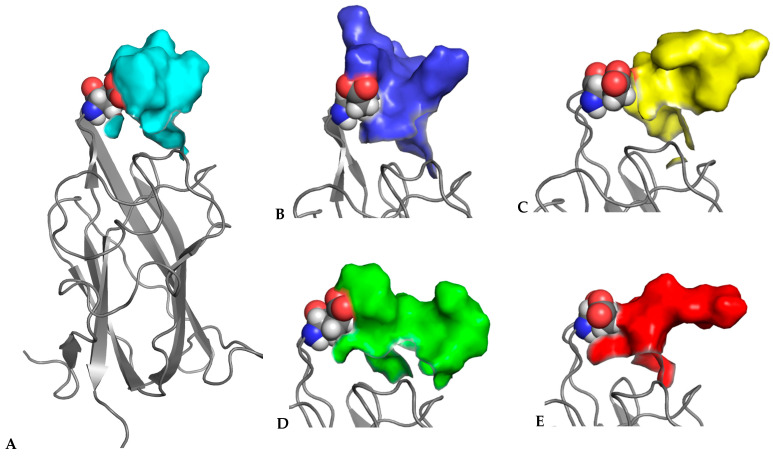
Representative structures of NRP1 in complex with peptides (**A**) C(cyan), (**B**) B167 (blue), (**C**) B186 (yellow), (**D**) A165 (green) and (**E**) control (red). Centroids of the most populated clusters for each of the complexes were extracted from trajectories of 30 ns molecular dynamics simulations, with NRP1 shown as a grey ribbon and Glu348 in CPK representation, while the peptides were shown as solvent-accessible surfaces. Differences in the molecular surfaces for each of the peptides will result from the combination of different backbone conformations and the size of the side chains within the sequences of the specific peptides.

**Figure 4 biomolecules-12-00372-f004:**
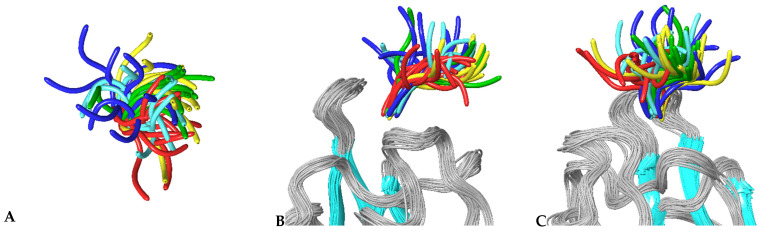
Sampled conformational space of NRP1-peptide complexes. Representative structures of the ten most populated clusters obtained from two repeats of 30 ns molecular dynamics simulations were ovelayed for each complex, (**A**) top view, (**B**) side view that corresponds to the orientation of the top view and (**C**) side view rotated by 90°. NRP1 is shown as ribbons with beta strands coloured blue and loops coloured grey. Peptides are denoted by using single colours: C—cyan; B167—blue; B186—yellow; A165—green and control red.

**Figure 5 biomolecules-12-00372-f005:**
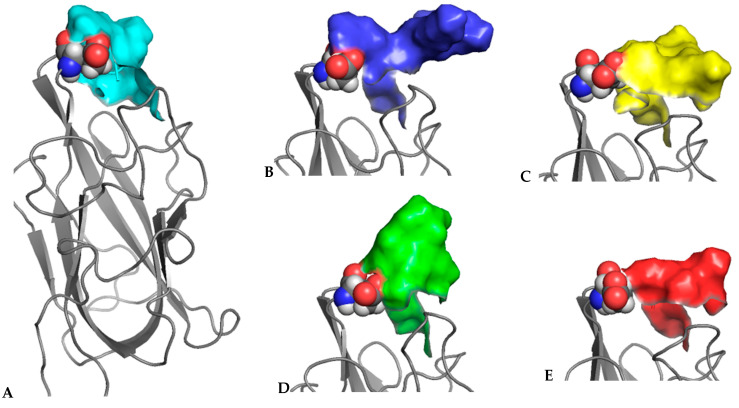
Representative structures of NRP2 in complex with peptides (**A**) C (cyan), (**B**) B167 (blue), (**C**) B186 (yellow), (**D**) A165 (green) and (**E**) control (red). Centroid of the most populated clusters were extracted from trajectories of 30 ns molecular dynamics simulations, with NRP2 shown as a grey ribbon and Glu351 in CPK representation, while the peptides were shown as solvent-accessible surfaces.

**Figure 6 biomolecules-12-00372-f006:**
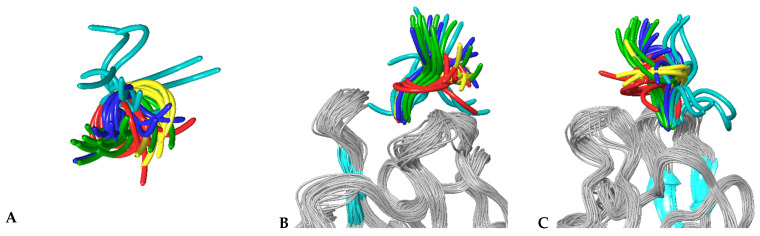
Sampled conformational space of NRP2 peptide complexes. Representative structure of ten most populated clusters obtained from two repeats of 30 ns molecular dynamics simulations were ovelayed for each complex, (**A**) top view, (**B**) side view that corresponds to the orientation of the top view and (**C**) side view rotated by 90°. NRP2 is shown as ribbons with beta strands coloured blue and loops coloured grey. Peptides are denoted by using single colours: C—cyan; B167—blue; B186—yellow; A165—green and control red.

**Figure 7 biomolecules-12-00372-f007:**
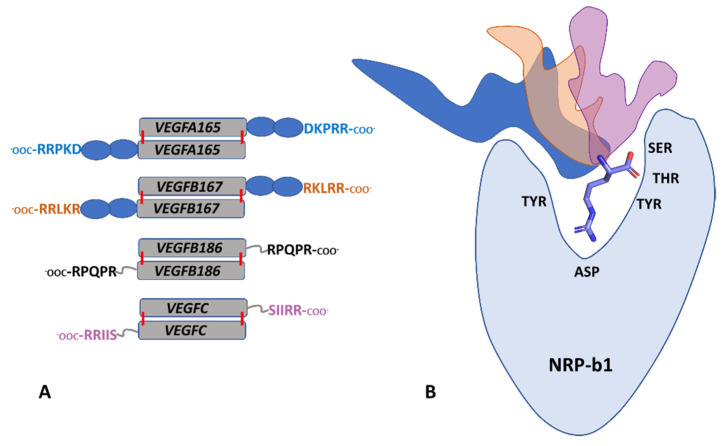
Different VEGF proteins use an overlapping binding site on the b1 domain of neuropilins. (**A**) Domain organization of VEGFA165, VEGFB167, VEGFB186 and VEGFC showing the C-terminal peptide sequences used in this study. The core cysteine-knot domains are colored grey, and the heparin-binding domains are shown in blue. Intermolecular disulfide bonds linking the cysteine-knot domains are illustrated by red lines. Both cysteine-knot domains and heparin-binding domains also contain intramolecular disulfide bonds, but for clarity, these have not been included in the diagram. (**B**) A schematic diagram illustrating how different peptides could be anchored into the NRP-b1 binding site by the C-terminal arginine (represented as sticks) but oriented differently above the molecular surface of the NRP1 or NRP2 b1 domains. NRP-b1 domain is represented in light blue, while the molecular surfaces of the various peptides are shown as dark blue, orange and pink for A165, B167 and C, respectively. Within the simplified representation of the NRP-b1 domain, starting from the left, the amino acid names indicate the relative positioning of TYR297/299, Asp320/323, Tyr353/356, Thr349/352 and Ser346/349 in the NRP1/NRP2 binding sites.

**Table 1 biomolecules-12-00372-t001:** Peptides used in the study. Ac denotes Acetyl-group.

Protein	Peptide Sequence	Name in the Manuscript
VEGFA165	Ac-DKPRR	A165
VEGFB167	Ac-RKLRR	B167
VEGFB186	Ac-RPQPR	B186
VEGFC	Ac-SIIRR	C
Control	Ac-AAAAR	Con

**Table 2 biomolecules-12-00372-t002:** Dissociation constants for interactions between the VEGF-derived peptides and the b1 domains of NRP1 and NRP2, as determined by surface plasmon resonance experiments. SE is a standard error for the K_D_ values.

Peptide	NRP1-b1 K_D_ (μM)	SE	NRP2-b1 K_D_ (μM)	SE
A165	7.87	0.22	30.90	0.93
B167	4.12	0.36	13.00	0.56
B186	10.00	0.29	24.80	0.82
C	33.20	1.30	27.10	0.96
Con	33.10	2.60	92.20	13.00

**Table 3 biomolecules-12-00372-t003:** Table of crystallographic data and model quality for structures produced in this study.

	NRP1-b1/B186			NRP2-b1/B167			NRP2-b1/C		
PDB Code	6TKK			6TDB			6TJT		
Data collection									
Space group	P 21 21 21			C 1 2 1			P 1 21 1		
Unit cell	38.88	39.98	97.61	107.36	140.12	74.36	38.24	76.54	63.36
	90	90	90	90	132.97	90	90	96.06	90
Wavelength, Å	0.976			0.977			0.976		
Resolution low, Å	48.81			70.06			63.00		
Resolution high, Å	1.06			2.45			1.31		
Outer resolution shell, Å	1.09–1.06			2.58–2.45			1.38–1.31		
CC half	0.999 (0.559) *			0.996 (0.738)			0.998 (0.905)		
Average I/sigma(I)	29.4 (12.3)			10.6 (2.3)			12.6 (2.3)		
Redundancy	5.5 (1.5)			3.4 (3.4)			3.1 (2.4)		
Completeness, %	80.2 (12.3) *			98.9 (99.7)			98.9 (97.2)		
No. unique reflections measured	55,937 (645)			29,170 (4278)			86,786 (12,438)		
Refinement									
Resolution low, Å	48.805			70.06			48.64		
Resolution high, Å	1.06			2.45			1.31		
Unique reflections work/free	53,084/2785			27,731/1428			82,426/4267		
R value work/free	0.1361/0.1495			0.1827/0.2497			0.1533/0.1790		
No. of atoms	1543			5471			2955		
Ramachandran plot, favoured/allowed/outliers, %	97.48/2.52/0.00			92.66/6.70/0.64			95.87/4.13/0.00		
RMSD bond lengths, Å	0.012			0.004			0.008		
RMSD bond angles, °	1.299			0.691			1.095		

Numbers in brackets refer to the values for the outer resolution shell. * Estimates of the overall resolution limit for all data sets were made based on the half-dataset correlation, CC(1/2). Data for 6TKK has 90.09% completeness at 1.10 Å.

**Table 4 biomolecules-12-00372-t004:** Interaction of peptides with NRP1 deconvoluted as fraction time of contact occurrence per protein residue during 30 ns production simulation.

		Fraction Time of Interaction Occurrence (%)				
Residue	Binding Mode	A165	B167	B186	C	Control
Y297		51	40	88	52	38
N300	WB	51		52	45	
W301		96	72	98	93	68
N313	WB	52	33	43	51	
T316		96	94	74	95	96
D320	BF	99	99	99	99	99
S346		98	81	99	97	63
K347	WB	32	60	47	33	30
E348		57	93		53	
	WB			58	52	
T349		47			52	
K351	WB			32		
Y353		35		67		

WB—water bridge; BF—bifurcated interaction with two groups in the ligand.

**Table 5 biomolecules-12-00372-t005:** Interaction of peptides with NRP2 deconvoluted as a fraction time of contact occurrence per protein residue during 30 ns production simulation.

		Fraction Time of Interaction Occurrence (%)				
Residue	Binding Mode	A165	B167	B186	C	Control
Y299		31	64	86	32	85
D301	WB	88	100	97		93
R303				30	100	
	WB		42			93
W304		98	99	78	41	98
T319		96	75	80	92	71
D323	BF	99	99	99	99	98
S349		89	74	67	67	54
R350	WB	61				
E351		37	74	63	55	
	BF	48	30	30		
T352		56	99	78	81	60
Y356		99	98	94	99	
I418		57			50	

## Data Availability

The structures and the X-ray diffraction data were deposited to the PDB with accession numbers PDB ID: 6TKK, 6TDB, 6TJT.

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
