# Peer review of "VEGFA, B, C: Implications of the C-Terminal Sequence Variations for the Interaction with Neuropilins"

_biomolecules, 2022, doi:10.3390/biom12030372_

Round 1

Reviewer 1 Report

 Each isoform from the VEGF protein family employs a core domain for the specific interaction with a cognate tyrosine kinase receptor (VEGF-R), whereas their C-terminal sequence, and in particular the C-terminal arginine, interacts with the b1 domain of neuropilins (NRP). NRP1 is a primary receptor for VEGFA and VEGFB, while NRP2 is the main physiological receptor for VEGFC.

To decipher the contribution made by the interaction of the C-terminal sequences of the individual VEGF proteins with NRPs that leads to functional differentiation, the crystal structures of complexes for various NRP/VEGF-derived peptides pairs were determined and binding interaction was examined using surface plasmon resonance (SPR). It was shown by SPR that interaction with VEGF-derived pentapeptides increases thermostability of b1 domains of NRP1 and NRP2. However, the binding constants determined from SPR showed minor differences, not highlighting variations in the interactions of the various NRP/peptide complexes. Moreover, the crystal structures were solved for three complexes: NRP1-b1/VEGFB186, NRP2-b1/VEGFB167, NRP2-b1/VEGFC. Yet, these structures did not reveal significant differences in the binding mode of the peptide to the VEGF protein, and showed that only the last C-terminal residues were ordered and contributed to the interaction. However, molecular dynamic (MD) simulations showed that the flexibility of the peptides and their binding area vary significantly. Intriguingly, neither peptide B167 nor peptide C formed significant interactions with Tyr353 of NRP1. Moreover, peptide C, the main physiological substrate of NRP2 interacted with Arg303 but not Asp301, in contrast to peptides A165 and B167.

It was concluded that molecular dynamic simulation uncovered conformational preferences of the VEGF-derived peptides that contribute to the ligand selectivity of neuropilins. However, it is not clear to me why the interactions/non-interactions with Arg303, Asp301, Tyr353 explain the selectivity. This section, which is the main conclusion of the paper, should be deepened and figures of the corresponding structures shown to clarify this important point.

Major modifications

Line 187. How do you explain that the control peptide caused a Tm shift of 3°C for NRP1-b1, only a little bit less than the Tm shift of 3.6° caused by peptides A165 and B186 ? Do you really think that the increase in Tm is specific?

Lines 198-199. This poorer selectivity of NRP2-b1 observed in the SPR analysis mirrored a lack of distinction of peptides with respect to their effect on Tm1 for NRP2. On the contrary, you said above that B186 had the most stabilizing effect, compared to the other peptides (Sup Figure 1B).

Line 201 “VEGFA exhibited the weakest affinity for VEGFC-derived peptide and the affinity was similar to that of the control peptide ??? «  rather « VEGFA exhibited the weakest affinity for the VEGFC-derived peptide and the control peptide ? 

Line 202 “again consistent with the results of the thermal shift assays ». Not really. The Tm shift is high for the control peptide.

Table 3. All statistics should also be given for the last resolution shell. In particular, I am worried about the completion for theNRP1-b1/B186 crystal in the last shell. 

Figure 1: What is the electron density. 2Fobs-Fcalc? An Fobs -Fcalc omit map (omitting the peptide) should presented in each case and the level (s) should be given.

Lines 246-247. “Namely, backbone atoms of the B186 peptide formed hydrogen bonds with the backbone atoms of the symmetry related NRP1 residues within the loop L2 (residues 320-322). » The interaction of the peptide at the interface of two NRP1 molecules deserves a Figure.

Lines 253-255 An omit map around the peptide location should be shown, with the four modeled residues to understand that the 5thresidue could not be modeled.

Line 255: what do you mean by “the main conformation”?

Lines 259-260 “the addition of the acetyl group at the N-terminal arginine created a potential for interaction with the ligand-binding site on the NRP1-b1 domain. » Not understood. How is the acetyl group expected to interact ?

Lines 293-295 “The two figures demonstrate that the peptide corresponding to the key physiological ligand, VEGFA165, occupies a much tighter conformationalcluster » This is not clear at all in Figure 3. The green area in Figure 3D (for A165) seems as large as the areas in the other panels. With this reasoning, the cyan area (Figure 3A) for peptide C, with poorest interaction with NRP1-b2 should be the largest, which does not appear to be the case. But you can say that the areas are different for the various peptides.

Lines 345-346 “In crystal structures of NRP2, Arg303 is engaged in an internal ionic interaction with Asp315 which is maintained even when a peptidic ligand is present. » It would be nice to have a figure showing this region for the crystal structures of NRP2-b1/B167 and NRP2-b1/C complexes to understand the potential interactions of the peptides with Arg303 and Asp315. If this is an important result of the MD simulation that explains the differences of interactions for the various differences between the various NRP/VEGF complexes, this section should be developed further and conclusions stated more clearly.

The discussion section is poorly related to the MD data that provide the most important results about the differences between the various NRP/VEGF-derived peptides pairs examined.

Line 405 The discussion about the formation of the ternary complexes is purely speculative. Change « This molecular mechanism not only explains » to This molecular mechanism could not only explain... but also support»

Minor corrections 

Line 18 VEGFC

Line 42 distinct roles

Line 47 delete one « were »

Line 38 and 63 define NRP1

Line 75 neuropilin

Lines 75-76 interactions between NRP1 or NRP2 (NRP1/2) and the C-terminus of VEGFA have been described for a heterocomplex or in the context of a crystal structure of a fusion protein

Sentence not clear. Moreover, indicate which experiments showed the interaction and explain fusion protein (of which proteins exactly) ?

Line 80 theso-called « C-end rule »

Lines 90-91 sentence difficult to understand. Put comas, delete « after » ? suggestion : that, within the contextof the full structure of the key physiological molecule VEGFA165, comprise the disulphidebridged cysteine knot domain followed by the five C-terminal residues.

Line 96 change « postulate » by « conclude »

Line 107 « Tsai et al » add reference 22

Define « Ac » in Table 1 and line 110

Table 1 change NRP1b1 and NRP2b1 to NRP1-b1 and NRP2-b1

Line 178 what do you mean by « apo»-form of NRP1-b1 ? NRP1-B1 alone (without peptide)? Then it should be also the apo form for NRP2-B1 line 179.

Line 181 delete « the » in « the shifts »; what is NRP2b2 ??? NRP2-1 ?

Line 184 dot is missing. Tm2 of NRP2-b1.For

Table 2 define SE. no unit?

Supp Figure 2: The text, the x and y axes and the figures are not readable.

Table 2. Put the Kd figures in µM. It will be more easily readable. Same for SE figures

Line 197 delete “with a narrower range of dissociation constants » (it is the same as lower selectivity) and add also : which alsobound peptides with lesser affinity

Table 3 : NRP1-VEGFB186, NRP2-VEGFB167, NRP2-VEGFC should be named NRP1-b1/B186, NRP2-b1/B167, NRP2-b1/C, like in the text.

Lines 212-218. The crystallization conditions should be reported in the “Material and Method” section, not in the “Results” section.

Lines 212-218. Was any cryoprotectant used to freeze the crystals? On which beam line were the data collected?

Line 226 b-sheets

Line 227 with residues

Line 228 Residues S346/S349, D320/D332, E348/E351, T349/T352 are not shown in Figure 1. Either interactions with these residues are already known and it is not necessary to recall them in the text or a Figure that shows the labeled residues is needed. These residues could be shown in a supplementary Figure, in stereo if there are too many residues to include in the Figure.

Line 236. Delete “for all three complexes”

Line 240 delete “associated with the NRP2-b1 domain” change “while the two N-terminal residues of the pentapeptide were disordered. » To « the two N-terminal residues of the pentapeptide being disordered.

Lines 242 and 249 delete « in contrast »

Legend Figures 1 and 2   change NRP1b1 NRP2b1 to NRP1-b1 NRP2-b1

Figure 1. could you show the H-bond that is mentioned in the text lines 306-309 thas ssems so important : In all crystal structures of molecular complexes of NRP1 with its natural ligands, Tyr353 was identified 307 as one of the key residues forming a hydrogen bond with the carboxylate oxygen of ligand’s terminal arginine

Line 250 Figure 2A

Line 254 « while the remainder of the peptide displayeddisorder, and was not modeled. »

Figure 2A. add : The two peptides with only two modeled amino acids are at the top and the bottom whereas the peptide with four modeled amino acids is in the center.

Line 253 replace « and the 5th, N-terminally acetylated residue, displayed some disorder » by  « the 5th, N-terminally acetylated residue, being disordered »

Line 259 add B167 and the sequence: « the B167 peptide contains an almost palindromic sequence RKLRR »

Line 260 remove « for » : « a potential interaction »

Figure 2B : it should be indicated in Figure 2B or in the Legend of Figure where the N-terminal acetylated arginine (in molecule B) and the C-terminal arginine (in molecule A) groups are. Thr352B should be labeled.

Line 264. The N-acyl group is oriented towards Trp304. Is there a cation-pi interaction between the N-terminla arginine and Trp304 ? If yes, you should say it.

Lines 265-266 « and losing an interaction with Thr352 (Thr349 in NRP1) which is otherwise one of the hallmarks of the canonical binding mode for the C-terminal arginine. » You should first recall the normal binding mode : « The C-terminal arginine interacts with Ser349, Tyr356 and Asp ??… In contrast, the N-terminal acetylated arginine… »

Line 295 : « with the structures oriented near the surface of the NRP1-b1 molecule. » not understood

Lines 302-303 : replace « but it is the only peptide that has a strong preference for Glu348 and interacts with this residue 93% of the time » by « despite its strong preference for Glu348, as shown by the 93% time of interaction occurrence with this residue. »

Line 310 add Figure 1A. « In all crystal structures of molecular complexes of NRP1 with its natural ligands, Tyr353 was identified as one of the key residues forming a hydrogen bond with the carboxylate oxygen of the ligand’s terminal arginine while simultaneously engaged in stacking interactions with the aliphatic portion of the arginine side chain (Figure 1A)

Lines 333-337 change « Table 5 clearly demonstrates that for the case of NRP2-b1 domain control peptide 335 also exhibits fewer interactions overall and in contrast to what was observed for the NRP1-336 b1 domain, the control peptide does engage T352 (T349 in NRP1) but only 60% of the time,similarly to what is seen for A165. » to « Table 5 clearly showsthat, inthe case of NRP2-b1 domain,thecontrol peptide also exhibits fewer interactions overall compared to the other peptides, but that, andin contrast to what was observed for the NRP1b1 domain, the control peptideitdoes engage interactions withT352 (T349 in NRP1) but only 60% of the time,similarly to what is seen for A165. »

Line 338 change « for the full duration of 30 ns »  to « for the 30 ns simulation »

Title of 6th column Tables 4 and 5 why « C1 » and not « C » ?

Same for Legend figure 5

Line 344 change « In NRP2 L1 loop is one residue longer withthe sequence of YSDGRW compared to YSTNW in NRP1 » to « In NRP2, theL1 loop (YSDGRW)is one residue longer than that ofNRP1 (YSTNW)

Line 389 uncovered ?

Check the references. Sometimes the number of authors is trunctated to 3, sometimes not.

Reviewer 2 Report

The submitted by authors manuscript “VEGFA, B, C: implications of the C-terminal sequence variations for the interaction with neuropilins” to “Molecular Structure and Dynamics” ’s section of the journal BIOMOLECULES, complements authors group work, as the “Peptides Derived from Vascular Endothelial Growth Factor B Show Potent Binding to Neuropilin-1” and other, the last years.

They applied many experimental studies, like protein expression and purification, surface plasmon resonance, crystallography, thermostability shift assays, but the obtained data could not explain the noticed selectivity and ligand preference of NRP1.

Authors proceeded in Molecular Dynamics Simulations, using a number of tools, like the Simulation Interaction Diagram (SID) module of the Maestro program. Based on the received data, authors were able to propose an explanation of the neuropilin selectivity.

Authors have to explain this discrepancy between their experimental data using techniques, like crystallography, SPR, and data received by simulation.

What changes are needed in the experimental design to be able to receive the evidence needed for the “accuracy” of the simulation data?

They have to provide some additional experimental data.

Author Response

Please see the attachment which includes the response to Reviewer 1 and the Reviewer 2.

Round 2

Reviewer 1 Report

see included file

Reviewer 2 Report

The revised paper has been improved both in several points in text as well figures/legends were updated and now paper is accepted for publication

Author Response

We thank the Reviewer 2 for finding that the manuscript is now of the quality that is acceptable for publication.